# Limbic Expression of mRNA Coding for Chemoreceptors in Human Brain—Lessons from Brain Atlases

**DOI:** 10.3390/ijms22136858

**Published:** 2021-06-25

**Authors:** Fanny Gaudel, Gaëlle Guiraudie-Capraz, François Féron

**Affiliations:** Institute of Neurophysiopathology, Aix Marseille University, CNRS, INP, 13005 Marseille, France; fannygaudl@gmail.com (F.G.); francois.feron@univ-amu.fr (F.F.)

**Keywords:** olfactory receptors, pheromonal receptors, gustatory receptors, transcript expression, limbic system, aging

## Abstract

Animals strongly rely on chemical senses to uncover the outside world and adjust their behaviour. Chemical signals are perceived by facial sensitive chemosensors that can be clustered into three families, namely the gustatory (TASR), olfactory (OR, TAAR) and pheromonal (VNR, FPR) receptors. Over recent decades, chemoreceptors were identified in non-facial parts of the body, including the brain. In order to map chemoreceptors within the encephalon, we performed a study based on four brain atlases. The transcript expression of selected members of the three chemoreceptor families and their canonical partners was analysed in major areas of healthy and demented human brains. Genes encoding all studied chemoreceptors are transcribed in the central nervous system, particularly in the limbic system. RNA of their canonical transduction partners (G proteins, ion channels) are also observed in all studied brain areas, reinforcing the suggestion that cerebral chemoreceptors are functional. In addition, we noticed that: (i) bitterness-associated receptors display an enriched expression, (ii) the brain is equipped to sense trace amines and pheromonal cues and (iii) chemoreceptor RNA expression varies with age, but not dementia or brain trauma. Extensive studies are now required to further understand how the brain makes sense of endogenous chemicals.

## 1. Introduction

Chemical senses play a key role in major functions. They are essential for finding food, detecting mates and predators, recognizing territories and avoiding danger [1,2,3,4,5,6,7,8,9,10,11,12,13]. Canonically, a molecule or a cocktail of molecules binds to chemoreceptors (CRs) located in the nasal and oral areas and induces electrical signals that are decoded by the central nervous system [14,15]. However, the academic view of chemical senses, making only sense of the external world, has been broadened by the discovery of ectopic CRs at the turn of the 21st century [16]. In other words, chemical senses are exteroceptive as well as interoceptive senses—they could taste and smell the non-self and the self.

Taste and olfactory receptors (TASR and OR) have been found in nearly every assessed organ. In most cases, these ectopic receptors do not exert any function in interoception, defined as the sense of the internal state of the body [17,18], since the tissue that is hosting them is not connected to the nervous system. However, they are involved in physiological functions. Bitter taste receptors have been identified in human and mouse airway smooth muscles. They reduce airway obstruction in a mouse model of asthma by producing muscle relaxation when activated [19,20,21,22]. Specific ORs play a part in spermatozoid mobility [23,24,25], skin repair [26,27,28] and may work as sentinels in the lungs and the gastrointestinal tract [29,30]. It seems that the gut “tastes” parasites before initiating immune responses [31,32,33] and kidneys “smell” fatty acids and respond by regulating blood pressure [34].

CRs are identified in the central nervous system [35,36]. Transcripts of six TASRs and of 12 ORs are present in the human frontal cortex and their expression is modulated during neurodegenerative diseases (Alzheimer, progressive supranuclear palsy, Creutzfeldt-Jakob, Parkinson and schizophrenia) [37,38,39,40]. Our team reported that the expression of two OR genes—Olfr110/111 and Olfr544—is impaired in the brain of a mouse model of Alzheimer’s disease [41]. In vertebrates, OR genes are well conserved and share a high percentage of homology as they are phylogenetically derived from nine common ancestor genes. For example, OR5V1 and OR52K1/OR52K2 are potential human orthologs of Olfr110/111 and Olfr544, respectively, with over 80% shared identity. The initial focus of the current study was to describe the expression of the latter within the human brain. Taking advantage of the freely accessible brain atlases, we used the transcriptome data (cDNA microarray, RNA sequencing) to expand our scope of investigation to a wider range of chemoreceptors and chart the genetic expression of human cerebral CRs. Nonetheless, no study has ever exhaustively mapped CRs within the encephalon.

For this purpose, we assessed the expression of taste receptors (TAS1R, TAS2R), pheromone-associated receptors (VNR—vomeronasal receptors and FPR—formyl peptide receptors), pheromone/odour-associated receptors (TAAR—trace amine-associated receptors) and olfactory receptors (ORs) as well as their canonical partners (G proteins, ion channels and elements of the transduction pathway) in human cerebral areas.

## 2. Results

### 2.1. Human Chemoreceptor Transcripts Are Expressed in the Brain

The human face harbours numerous OR-associated genes (396 coding genes and 425 pseudogenes), a smaller number of TASR-associated genes (28 coding genes and 11 pseudogenes), a few number of TAARs (six coding genes and three pseudogenes) and a variety of pheromone receptor-associated genes (VNR — four coding genes, 200 pseudogenes and FPR — three coding genes and no pseudogene). The Allen Brain transcriptome reveals that most of the CR-related coding genes are expressed in the human brain. Figure 1 summarises the main findings: 95% (376/396) of OR genes and 100% of TASRs, TAARs, VNRs and FPRs genes are expressed. Conversely, the human brain excludes most if not all chemoreceptor-linked pseudogenes: only 2% of OR and 1% of VNR pseudogenes are detected in the central nervous system.

### 2.2. Expression of Chemoreceptors May Vary with Age but Not Brain Diseases

We compared the mean expression of the selected 10 class I and 10 class II ORs, three type 1 and 10 type 2 TASRs. Class II ORs are significantly more expressed than class I ORs (2.46 ± 0.04 vs. 1.99 ± 0.04, respectively) and TAS2Rs significantly more than TAS1Rs (4.28 ± 0.05 vs. 2.16 ± 0.05, respectively) (Figure 2A). To assess the extent of inter-individual variations, we compared the cerebral transcript expression of two CRs (OR2L13 and TAS2R14), including donors of different ages, genders and ethnic backgrounds. A relatively stable regional expression, between donors, of OR2L13, the most produced class II OR transcript, is observed (Figure 2B). Overall, OR2L13 expression reaches a zenith in cortices—frontal, parietal, occipital—and a nadir in amygdala, basal forebrain and hypothalamus (Figure 2B, top). Conversely, the regional expression of TAS2R14, the most expressed TAS2R, varies greatly between donors (Figure 2B, bottom). Apropos of age, variations are also observed. The overall cerebral expression of OR2L13—3.70 ± 0.16 (24 years), 4.56 ± 0.09 (31 years), 3.25 ± 0.09 (39 years), 4.11 ± 0.10 (55 years), 3.29 ± 0.10 (57 years)—and TAS2R14—6.13 ± 0.11 (24 years), 6.66 ± 0.08 (31 years), 6.04 ± 0.06 (39 years), 6.42 ± 0.05 (55 years), 6.14 ± 0.08 (57 years)—indicates that both CRs mRNA are produced in lower quantity in the eldest donor. However, this finding cannot be fully associated to a more advanced age since a low expression is also noticed in the 39 year-old brain. Expression of these two receptors was also compared between a woman and a man of similar ages. The overall cerebral expression of OR2L13—4.12 ± 0.10 (male), 3.53 ± 0.12 (female)—and TAS2R14—6.43 ± 0.05 (male), 6.14 ± 0.06 (female)—does not seem to indicate a sexual dimorphism in the expression of both genes. Similarly, the regional expression of OR2L13 and TAS2R14 was comparable between the male and the female subjects (Figure 2C).

The Allen brain database includes transcript values from patients with dementia and/or traumatic brain injury and individuals without known pathology. Statistically significant differences in transcript expression were observed when pathological or traumatized brains were compared to control individuals. However, no CR RNA was found to be significantly deregulated.

### 2.3. Gene Expression of Brain Chemoreceptors Is Confirmed by Other Databases

To avoid a possible bias related to the use of a single atlas, we assessed the transcript expression of our selected 20 olfactory and 13 taste receptors in three other databases: GTEx, BioGPS and Human Protein Atlas (Table 1 and Table 2). Although data are unavailable for some CRs, a more or less robust expression of our candidates is observed. Similarly to what has been reported above, OR2L13 and TAS2R14 are the most transcribed genes, according to GTEx. However, according to BioGPS, OR52K1 and TAS1R3 stand as the most expressed genes. In addition, according to the Human Protein Atlas, 74% (14,518) of all human protein coding genes (19,613) are expressed in the brain and about 10% (1460) of these genes show an elevated expression in brain compared to other tissue types. Among them, OR14I1 is classified as tissue enriched and OR2L13 as tissue enhanced (https://www.proteinatlas.org/humanproteome/tissue/brain, accessed in 2020).

Analysis of the data available in several collections such as BioGPS Gene Atlas and GTEx. BioGPS data, extracted from Affymetrix chips, are expressed as fluorescent intensities, after GCRMA (GC Robust Multi-Array Average) normalisation. Values from GTEx correspond to a number of transcripts per million.

Finally, we compared the transcript expression of brain CRs to lingual CRs. No significant difference was noticed between the two tissues (Appendix A).

### 2.4. ORs Are Primarily Expressed by the Limbic System

We performed a more detailed analysis of the Allen brain atlas data and focused on the regionalized CR distribution in the brain of a single individual (31 year-old male). Both families of ORs—class I, class II—are expressed with an average log2 expression of 1.99 ± 0.04 and 2.46 ± 0.04, respectively (Figure 3A,B,H). On average, class II ORs are significantly more expressed than class I ORs (****, *p* < 0.0001) (Figure 2A). ORs are predominantly expressed in the limbic system (amygdala, basal forebrain, basal ganglia, limbic cortex and hypothalamus). For the 10 selected class I ORs, their mean expression varies as follows: amygdala (2.32 ± 0.17), hypothalamus (2.16 ± 0.13), basal forebrain (2.13 ± 0.14), limbic cortex (2.05 ± 0.27), basal ganglia (2.01 ± 0.21), frontal cortex (1.96 ± 0.29), temporal cortex (1.95 ± 0.30), parietal cortex (1.94 ± 0.30), occipital cortex (1.91 ± 0.29), mesencephalon (1.91 ± 0.22), thalamus (1.86 ± 0.19), myelencephalon (1.83 ± 0.25) and metencephalon (1.82 ± 0.22) (Figure 3A).

For the 10 selected class II ORs, the following mean expression is observed: amygdala (2.72 ± 0.24), basal forebrain (2.63 ± 0.27), hypothalamus (2.60 ± 0.26), basal ganglia (2.52 ± 0.27), myelencephalon (2.49 ± 0.35), limbic cortex (2.48 ± 0.31), thalamus (2.45 ± 0.34), metencephalon (2.42 ± 0.34), occipital cortex (2.41 ± 0.35), parietal cortex (2.34 ± 0.32), mesencephalon (2.34 ± 0.33), temporal cortex (2.34 ± 0.31) and frontal cortex (2.28 ± 0.32) (Figure 3B).

Within these two subfamilies, OR52K2 (3.82 ± 0.17) and OR2L13 (4.56 ± 0.09) stand out as the most expressed class I and class II transcripts in all studied areas (Figure 3H). Of note, within the amygdala, the strongest expression is associated to OR10A2. (Appendix A, top).

In order to assess the most highly expressed olfactory receptors, we compared the number of copies in the four nuclei of the amygdala (Appendix A). The top seven ORs (log2 value > 7) within the amygdala are: OR2H1, OR3A1, OR4D2, OR5L2, OR10C1, OR10H2 and OR13A1. However, it should be noted that some discordant values between probes are sometimes observed. In addition, inter-individual variability was assessed by comparing the level of expression of all ORs within the amygdala of the six individuals (five men and one woman; age range: 24–55 years) included in the Allen brain atlas cohort (Appendix A). For the same receptor in the same nucleus, expression variations can range from simple to quadruple. However, no effect of age was observed.

### 2.5. TAS2Rs Display a Higher Expression

Both type 1 (sweet- and umami-sensing) and type 2 (bitter-sensing) TASRs are expressed in the human brain, with a mean expression of 2.16 ± 0.05 and 4.28 ± 0.05, respectively (Figure 3C,D,H). TAS2Rs are significantly more expressed than TAS1Rs (Figure 2A, ****, *p* < 0.0001).

TASRs are predominantly expressed in the limbic system. The mean expression of TAS1Rs varies according to the brain area, as follows: basal forebrain (2.50 ± 0.44), amygdala (2.42 ± 0.41), hypothalamus (2.39 ± 0.41), basal ganglia (2.25 ± 0.55), mesencephalon (2.21 ± 0.54), metencephalon (2.13 ± 0.45), limbic cortex (2.11 ± 0.47), frontal cortex (2.04 ± 0.40), parietal cortex (2.04 ± 0.44), occipital cortex (2.03 ± 0.39), temporal cortex (2.01 ± 0.44), myelencephalon (1.99 ± 0.46) and thalamus (1.94 ± 0.36) (Figure 3C). For the TAS2Rs, the following mean expression is noticed: basal ganglia (4.53 ± 0.55), amygdala (4.49 ± 0.51), myelencephalon (4.46 ± 0.62), basal forebrain (4.45 ± 0.52), mesencephalon (4.39 ± 0.60), occipital cortex (4.36 ± 0.57), metencephalon (4.30 ± 0.59), hypothalamus (4.21 ± 0.50), thalamus (4.20 ± 0.56), parietal cortex (4.18 ± 0.53), limbic cortex (4.13 ± 0.54), temporal cortex (3.96 ± 0.51) and frontal cortex (3.95 ± 0.52) (Figure 3D).

TAS1R1 and TAS2R14 stand out as the most expressed transcripts (2.97 ± 0.07 and 6.66 ± 0.08, respectively), when averaging all studied areas (Figure 3H). Nonetheless, the most expressed TAS2Rs vary according to the observed brain area. For example, TAS2R31 displays the highest expression in the basal forebrain and hypothalamus (Appendix A, bottom).

In addition, we describe the cerebral expression of four genes encoding candidates of sour perception—ASIC2, KCNK3 (also known as TASK-1), PKD2L1 and OTOP1. All are observed as expressed at the transcriptomic level in the human brain, with an average expression of 3.95 ± 0.20 (ASIC2: 6.55 ± 0.21; KCNK3: 4.69 ± 0.24; PKD2L1: 2.91 ± 0.69; OTOP1: 1.65 ± 0.10) (Figure 4A). They are predominantly expressed in the cortex and the hypothalamus: temporal cortex (4.67 ± 1.16), frontal cortex (4.67 ± 1.16), parietal cortex (4.56 ± 1.09), occipital cortex (4.22 ± 1.00), limbic cortex (4.17 ± 1.11), hypothalamus (3.80 ± 1.19), basal forebrain (3.74 ± 1.16), myelencephalon (3.71 ± 1.33), metencephalon (3.70 ± 1.23), mesencephalon (3.67 ± 1.24), amygdala (3.62 ± 0.98), thalamus (3.52 ± 1.18) and basal ganglia (3.32 ± 0.90) (Figure 4B).

Among these sour-associated genes, ASIC2 reaches an acme with a mean log2 expression of 6.55 ± 0.21 in the whole brain (Figure 4A).

### 2.6. The Human Brain Is Capable of Sensing Trace Amines and Pheromonal Cues

Trace amine and pheromonal cues are deciphered by three CR families—TAARs, VNRs, and FPRs—that are all present in the human brain, with an average log2 expression of 2.52 ± 0.06, 2.26 ± 0.04, and 2.85 ± 0.05, respectively. Their cerebral area-related distribution is as follows.

TAARs: amygdala (3.02 ± 0.49), basal forebrain (2.72 ± 0.43), hypothalamus (2.66 ± 0.43), basal ganglia (2.64 ± 0.44), mesencephalon (2.52 ± 0.54), myelencephalon (2.52 ± 0.49), limbic cortex (2.47 ± 0.47), metencephalon (2.43 ± 0.45), occipital cortex (2.39 ± 0.41), thalamus (2.38 ± 0.46), temporal cortex (2.38 ± 0.44), parietal cortex (2.34 ± 0.42) and frontal cortex (2.27 ± 0.41) (Figure 3E).

VNRs: amygdala (2.50 ± 0.57), basal forebrain (2.48 ± 0.61), basal ganglia (2.36 ± 0.70), hypothalamus (2.31 ± 0.54), metencephalon (2.29 ± 0.56), limbic cortex (2.25 ± 0.68), occipital cortex (2.24 ± 0.78), myelencephalon (2.23 ± 0.59), parietal cortex (2.18 ± 0.78), frontal cortex (2.17 ± 0.74), thalamus (2.16 ± 0.55), temporal cortex (2.16 ± 0.76) and mesencephalon (2.07 ± 0.53) (Figure 3F).

FPRs: hypothalamus (3.15 ± 0.90), basal forebrain (3.07 ± 0.98), amygdala (3.06 ± 0.73), mesencephalon (2.97 ± 1.16), basal ganglia (2.29 ± 0.85), myelencephalon (2.92 ± 1.16), metencephalon (2.83 ± 1.07), limbic cortex (2.79 ± 0.90), thalamus (2.77 ± 1.07), temporal cortex (2.68 ± 0.79), parietal cortex (2.60 ± 0.79), occipital cortex (2.64 ± 0.80) and frontal cortex (2.57 ± 0.73) (Figure 3G).

Within these three trace amine- and pheromone-associated receptors, TAAR5, VN1R1 and FPR1 stand out as the most expressed transcripts with a mean log2 score of 4.03 ± 0.06, 4.18 ± 0.09, and 4.65 ± 0.12, respectively (Figure 3H).

### 2.7. Canonical Chemoreceptor Partners Are Expressed in the Brain

With the exception of sour perception-associated ion channels, CRs belong to the G Protein-Coupled Receptor (GPCR) superfamily and mostly partner with Gαs, Gαq and Gαi/o G protein families. Gαq family is the most widely expressed with an average log2 score of 7.30 ± 0.05, followed by the Gαi/o (6.88 ± 0.05) and Gαs (5.97 ± 0.12) families (Figure 5G).

Their cerebral area-related distribution is as follows:

Gαs: amygdala (6.67 ± 2.19), basal forebrain (6.54 ± 2.20), hypothalamus (6.38 ± 2.35), myelencephalon (6.25 ± 2.42), mesencephalon (6.15 ± 2.43), basal ganglia (6.14 ± 2.24), metencephalon (5.97 ± 2.30), thalamus (5.85 ± 2.22), limbic cortex (5.67 ± 2.05), occipital cortex (5.55 ± 1.96), parietal cortex (5.50 ± 1.96), frontal cortex (5.46 ± 1.85) and temporal cortex (5.44 ± 1.92) (Figure 3A).

Gαq: amygdala (7.58 ± 0.21), limbic cortex (7.51 ± 0.39), mesencephalon (7.41 ± 0.25), metencephalon (7.39 ± 0.42), myelencephalon (7.38 ± 0.27), basal ganglia (7.35 ± 0.29), parietal cortex (7.28 ± 0.68), thalamus (7.28 ± 0.36), basal forebrain (7.26 ± 0.10), occipital cortex (7.26 ± 0.69), frontal cortex (7.17 ± 0.67), temporal cortex (7.14 ± 0.63) and hypothalamus (6.92 ± 0.21) (Figure 3B).

Gαi/o: mesencephalon (7.21 ± 0.43), metencephalon (7.16 ± 0.44), myelencephalon (7.14 ± 0.43), parietal cortex (6.95 ± 0.39), basal ganglia (6.87 ± 0.35), thalamus (6.85 ± 0.39), basal forebrain (6.84 ± 0.44), frontal cortex (6.84 ± 0.41), temporal cortex (6.82 ± 0.39), occipital cortex (6.79 ± 0.41), limbic cortex (6.66 ± 0.46), hypothalamus (6.73 ± 0.46) and amygdala (6.56 ± 0.42) (Figure 3C).

Within each G protein family, GNAS, GNAQ, and GNAI1 are the most expressed transcripts, reaching a log2 score of 8.64 ± 0.19, 7.58 ± 0.11, and 8.06 ± 0.07, respectively (Figure 4G). GNAL is predominantly expressed in the amygdala, basal ganglia and thalamus, while GNA11 expression is salient in the amygdala, basal forebrain, basal ganglia, and hypothalamus (Appendix A).

### 2.8. Canonical Chemoreception Transducers Are Expressed in the Brain

TRP and CNG channels display an average log2 expression of 4.14 ± 0.07 and 1.77 ± 0.06 (Figure 5G). Their cerebral area-related distribution is as follows:

TRP: metencephalon (4.57 ± 1.86), myelencephalon (4.47 ± 1.82), mesencephalon (4.47 ± 1.82), thalamus (4.30 ± 1.76), hypothalamus (4.20 ± 1.71), basal forebrain (4.10 ± 1.68), amygdala (4.02 ± 1.64), temporal cortex (4.01 ± 1.64), frontal cortex (3.98 ± 1.63), parietal cortex (3.98 ± 1.62), occipital cortex (3.95 ± 1.61), limbic cortex (3.92 ± 1.60) and basal ganglia (3.90 ± 1.59) (Figure 5D).

CNG: amygdala (2.26 ± 1.60), basal forebrain (2.01 ± 1.42), hypothalamus (1.89 ± 1.34), occipital cortex (1.87 ± 1.32), temporal cortex (1.77 ± 1.25), frontal cortex (1.77 ± 1.25), basal ganglia (1.72 ± 1.22), parietal cortex (1.72 ± 1.22), limbic cortex (1.70 ± 1.20), metencephalon (1.60 ± 1.13), mesencephalon (1.56 ± 1.10), myelencephalon (1.55 ± 1.10) and thalamus (1.53 ± 1.08) (Figure 5E). TRPC1 and CNGA2 are the most expressed transcripts, reaching a log2 score of 8.39 ± 0.07 and 1.54 ± 0.05 (Figure 5G).

A partner of the olfactory transduction pathway (olfactory marker protein, OMP) is expressed in the brain, with an average log2 expression of 1.89 ± 0.04. Its cerebral distribution is as follows: hypothalamus (2.12 ± 0.08), myelencephalon (2.04 ± 0.44), basal forebrain (2.03 ± 0.12), parietal cortex (2.01 ± 0.13), basal ganglia (1.99 ± 0.12), amygdala (1.92 ± 0.16), temporal cortex (1.85 ± 0.04), limbic cortex (1.82 ± 0.07), frontal cortex (1.81 ± 0.06), mesencephalon (1.76 ± 0.14), occipital cortex (1.75 ± 0.04), metencephalon (1.74 ± 0.12) and thalamus (1.68 ± 0.09) (Figure 5F,G).

## 3. Discussion

This study is one of the very first to assess and quantitate the transcript expression of all known CR families within the human brain. The expression of these chemosensor families within the central nervous system comes as no surprise since they were described in a pioneer study led by using the next-generation sequencing tool [42]. However, this latter description was done in whole/undetermined samples of brain, occluding potential variation of gene expression related to cerebral areas.

The cross examination of the data leads to the following conclusions: (i) most genes encoding for CRs–TASRs, TAARs, VNRs and FPRs are expressed in the brain at a rather low level, (ii) ORs, TASRs, TAARs and VNRs are mostly expressed in the limbic system, (iii) canonical CR-associated partners and transducer genes are transcribed in the brain, (iv) inter-individual variations are observed for some CRs, but no significant difference in CR expression is found when demented or traumatized brains are compared to control individuals.

Previous studies indicate that some CRs are expressed in the central nervous system. Several ORs and TASRs transcripts are expressed in the human entorhinal, frontal, prefrontal cortices as well as substantia nigra and cerebellum [37,38,39,40,43,44,45]. With respect to trace amine and pheromone receptors, TAAR1 is involved in the regulation of dopaminergic neurotransmission in rodents and primates [46], while human astrocytes and microglial cells express FPR1 and FPR2 [47]. We show here for the first time that 95% of OR transcripts (376 out of 396) and 100% of other CR mRNAs—TASRs, TAARs, VNRs, FPRs—are expressed in all studied human brain areas. Interestingly, even a few OR and VNR pseudogenes are transcribed in the human brain (respectively, 10 and two, out of 425 and 200 in the whole human genome), a finding in line with previous studies [42,48,49]. Although they are unable to produce fully functional proteins, pseudogenes can play diverse roles. Pseudogene-bound transcripts regulate the expression of their non-pseudogenic counterparts in human cancer cells [50] and, in Drosophila, pseudogene-derived ORs bind to proteins with a tissue-dependent expression [51]. It can be surmised that the 10 OR and two VNR pseudogenes transcribed in the human brain play a regulatory role in gene expression. Nevertheless, further studies are required to determine whether these pseudogene-derived CR transcripts are involved in regulatory functions or translated into functional proteins.

Overall, ectopic CR transcripts are poorly expressed [42]. More specifically, as reported in our previous studies, cerebral mRNAs coding for two ORs display a multi-fold diminished expression when compared to those produced in the olfactory mucosa [41]. The current study confirms this finding. Nonetheless, TASR transcript levels are similar in the brain and the tongue. However, mean global values could be misleading since the distribution of the various neural cell types varies considerably from one tissue to another. For instance, we showed that Olfr110/111 and Olfr544 proteins are mostly expressed by cerebral neurons that are outnumbered by glial cells, a feature not observed in the nasal cavity [41]. Similarly, the TAAR family, which plays a role in cerebral physiology and pathology [52], is expressed by a minority of neural cells, as demonstrated for TAAR1 [53]. Still, a detailed study on cell type-associated expression is required to further assess the roles of cerebral CRs.

None of the brain areas under investigation in this study is devoid of CR transcripts. Their expression is most prominent in the limbic system and the brainstem. This finding does not come up as a major surprise since chemical senses, appearing before physical senses during evolution, are strongly associated to the primal and paleo-mammalian brains. The olfactory mucosa, which harbours olfactory and pheromone receptors, is linked to the limbic system via the olfactory bulb and the piriform cortex [54]. The tongue and its taste buds convey chemical information to the nucleus of the solitary tract of the brainstem and, then, the amygdala, thalamus and hypothalamus, through the VII^th^, IX^th^ and X^th^ cranial nerves [54]. Olfaction and taste are of major importance to lead behaviour and survival functions. Therefore, it is tempting to assume that cerebral CRs play a role in self-preservation and several limbic system-associated functions like memory and regulation of autonomic and endocrine metabolisms, in response to emotional stimuli.

In support of this association between brain CRs and limbic system, it can be noticed that the pattern of expression of Gαs—canonically involved, via Gαolf, in the OR and TAAR signalling cascades [55,56]—is very close to CR mapping. This almost perfect juxtaposition between CRs and Gαs is, however, not true for all the second messengers. FPRs and VNRs are known to be associated to the Gαi/o family [57,58,59], but they are more consistently expressed in mesencephalic structures and the hindbrain (metencephalon and myelencephalon) than in the limbic system. Likewise, the expression of TRPs and other G proteins is ubiquitous, an unsurprising finding since these effectors are associated to many receptors not linked to chemo-perception. In respect of functionality, it can be noticed that the coupling of CRs to G proteins varies and ORs can be coupled with Gαs, Gαolf and Gα15/16 proteins [60]. In sensory olfactory neurons, the OR-Gαolf coupling ensures the recognition of odours and the local sorting of axons, while the OR-Gαs association promotes axon targeting of the appropriate glomerulus within the olfactory bulb, during maturation [61].

As demonstrated through the examples of OR2L13 and TAS2R14, patterns of expression of brain CRs can be similar or vary, in an age- and individual-specific manner. Nevertheless, the size of our cohort is too small to draw any valid conclusion. In parallel, previous studies indicated that the expression of some cerebral ORs and TASRs is modulated in patients with neuropathologies [37,38,39,40]. We assessed CR expression in brains of individuals with dementia or after a traumatic brain injury. Intriguingly, no significant variation in CR expression was observed. Such a discordant finding indicates that additional studies are required to enlighten this issue. However, it can be pointed out that the previous study on OR expression in the frontal and entorhinal cortices was performed with the PCR technique (38) while the Allen brain atlas team used cDNA microarrays.

Although exhaustive, this study is hampered by limitations. We studied all members of CR families but limited our analysis to 10 genes. Investigating the whole OR and TASR families would provide a more accurate information. A databank quantifying the level of expression of brain proteins is lacking. However, there is little doubt that CRs are translated into proteins within the central nervous system, as multiple studies conclusively demonstrated the presence of olfactory and taste receptors in several brain areas [37,38,39,40,41,62,63,64,65,66,67].

In summary, we performed a comprehensive analysis of transcript expression of CRs within the human brain using freely accessible genomic expression databases. All members of the TASR, TAAR, VNR and FPR families, as well as 95% of the OR family, are expressed in the brain. This study reveals that CR genes are mainly expressed in the limbic system, underlying the potential importance of CRs in self-preservation. Their expression patterns seem to be gene-dependent in the different brain zones, which could indicate that they have different cerebral functions. We found that CR canonical transducers are expressed in these areas, supporting the idea that CRs may be functional in the brain. This work suggests that CRs are expressed in the brain and could be involved in unknown physiological and pathological cellular mechanisms. They could act as biosensors to detect pathological states in the brain and trigger the appropriate response. Nevertheless, to be (un)validated, these hypotheses require additional comprehensive studies. Authors should discuss the results and how they can be interpreted from the perspective of previous studies and of the working hypotheses. The findings and their implications should be discussed in the broadest context possible. Future research directions may also be highlighted.

## 4. Materials and Methods

### 4.1. Gene Selection

Human CR family includes 376 ORs [68], clustered in class I and class II ORs. We limited our study to 10 class I, and 10 class II ORs. OR2D2, OR2L13, OR2T1, OR2T33, OR4F4, OR6F1, OR10G8, OR11H1, OR51E1, OR52H1, OR52L1 and OR52M1 were documented as expressed in the human brain [38,39]. OR5V1 shares 81% identity with Olfr110/111. OR52K1 and OR52K2 share 82% and 80% identity, respectively, with Olfr544. Both Olfr110/111 and 544 genes are expressed in the mouse brain [38,41]. OR10A2, OR51A4, OR51A7, OR52B6 and OR52E8 were randomly selected to reach a total of 10 ORs per class. TASRs are classified in type 1 (TAS1R), which includes three members TAS1R1, TAS1R2 and TAS1R3 [69], and type 2 (TAS2R) that comprises 25 members [70]. TAS2R4, TAS2R5, TAS2R10, TAS2R13, TAS2R14 and TAS2R50 are found in the human brain [38,39]. Other TAS2R (TAS2R1, TAS2R31, TAS2R38 and TAS2R45) were randomly selected to reach a total of 10 TAS2R (Table 3). We report the expression of VN1R1, VN1R2, VN1R4 and VN1R5 in the human brain. The cerebral expression of the six functional TAARs–TAAR1, TAAR2, TAAR5, TAAR6, TAAR8, TAAR9– and the 3 FPRs–FPR1, FPR2, FPR3–was quantified. We included the cerebral expression of described potential sour chemosensors: ASIC2 [71], KCNK3 [72], PKD2L1 [73,74], and OTOP1 [75]. We explored the brain expression of several CR-associated G proteins: Gαs family (GNAL, GNAT3, GNAS), Gαq family (GNAQ, GNA11), Gαi/o family (GNAI1, GNAI2, GNAI3, GNAO) and partners (CNGA2, CNGA4, OMP, TRPC1, TRPM4, TRPM5, TRPV2, TRPV4).

### 4.2. The Allen Human Brain Gene Expression Analysis

Most of the data reported were obtained from the Allen Brain Institute’s (https://www.brain-map.org, accessed in 2021). The Allen Human Brain Atlas (http://human.brain-map.org, accessed in 2021) includes RNA microarray data collected from post-mortem brains of 6 donors, with no known neuropsychiatric or neuropathological history as described in ALLEN Human Brain Atlas Normalization, Microarray Data (2013).

We analysed CR expression levels in all six donors (five men, one woman) as well as in the brains of individuals with dementia (*n* = 50) and/or traumatic brain injury (*n* = 50) and individuals with no known pathology (*n* = 50). Cerebral expression of selected CRs and CR-partners was assessed using microarray data available on the Allen Institute’s Allen Brain Atlas (http://human.brain-map.org/microarray/search, accessed in 2021). Details are available on the Allen Institute’s website (http://help.brain-map.org/display/humanbrain/documentation, accessed in 2021). Briefly, the log2 level of expression was collected for all genes and associated to each brain area mapped (97 areas, Figure 6). Different probes were used to detect the genes of interest, allowing to average the log2 levels and the results were compiled. To compare the level of expression of each CR family, we used a commonly expressed receptor in the central nervous system, NMDAR2B, as an internal positive control.

### 4.3. Gene Expression Analysis Using other Databases

To (in)validate the Allen Brain Atlas-associated findings, we analysed the data available in several collections: BioGPS Gene Atlas (http://biogps.org/, accessed in 2020), GTEx (https://gtexportal.org/home/, accessed in 2020) and Human Protein Atlas (https://www.proteinatlas.org, accessed in 2020). BioGPS data (Affymetrix chips) are expressed as fluorescent intensities, after GC Robust Multi-array Average normalisation. GTEx and Human Protein Atlas values correspond to a number of transcripts per million.

### 4.4. Statistical Analysis and Visuals

All data are presented as means ± SEM and were analysed using GraphPad Prism6 software. Statistical analyses were performed using non-parametric one-way ANOVAs (Kruskal–Wallis) and the post-hoc Dunn’s multiple comparison test. The Mann–Whitney test was used for significance between groups presented in Appendix A. Differences between mean values were considered statistically significant when *p* < 0.05 (*), *p* < 0.01(**), *p* < 0.001(***), *p* < 0.0001 (****). Visuals were made using GraphPad Prism6, Inkscape and Adobe Photoshop software.

## Figures and Tables

**Figure 1 ijms-22-06858-f001:**
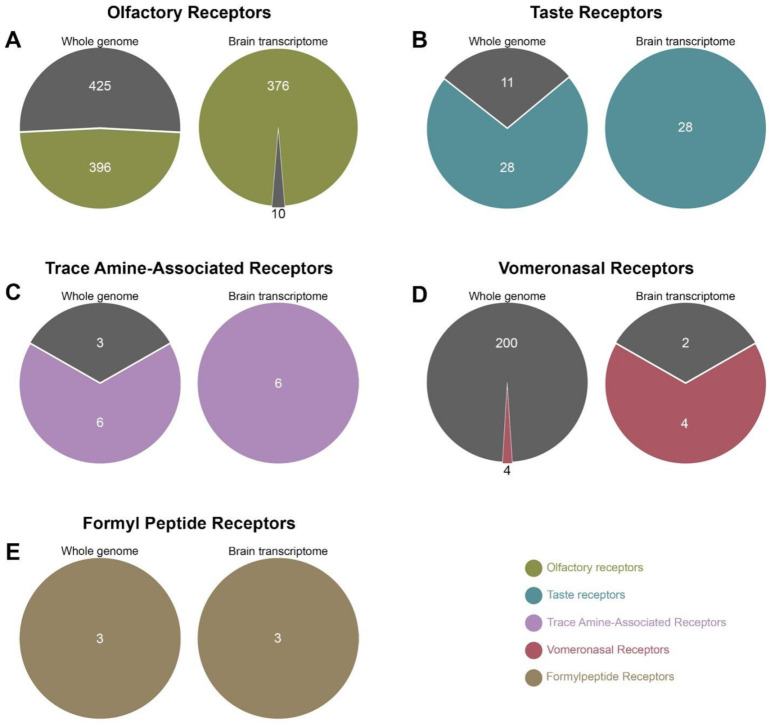
Genome versus transcriptome comparison of human chemoreceptors (CRs). The number of genes and pseudogenes expressed in the whole genome (left pie charts) was compared to those found to be transcribed (right pie charts), for all CR families: olfactory receptors (**A**, green), taste receptors (**B**, blue), trace amine-associated receptors (**C**, purple), vomeronasal receptors (**D**, red) and formyl peptide receptors (**E**, brown), The coloured areas correspond to coding genes and dark grey areas to pseudogenes. The census of genes expressed in the brain transcriptome was carried out using data from all five subjects available in the Allen brain atlas.

**Figure 2 ijms-22-06858-f002:**
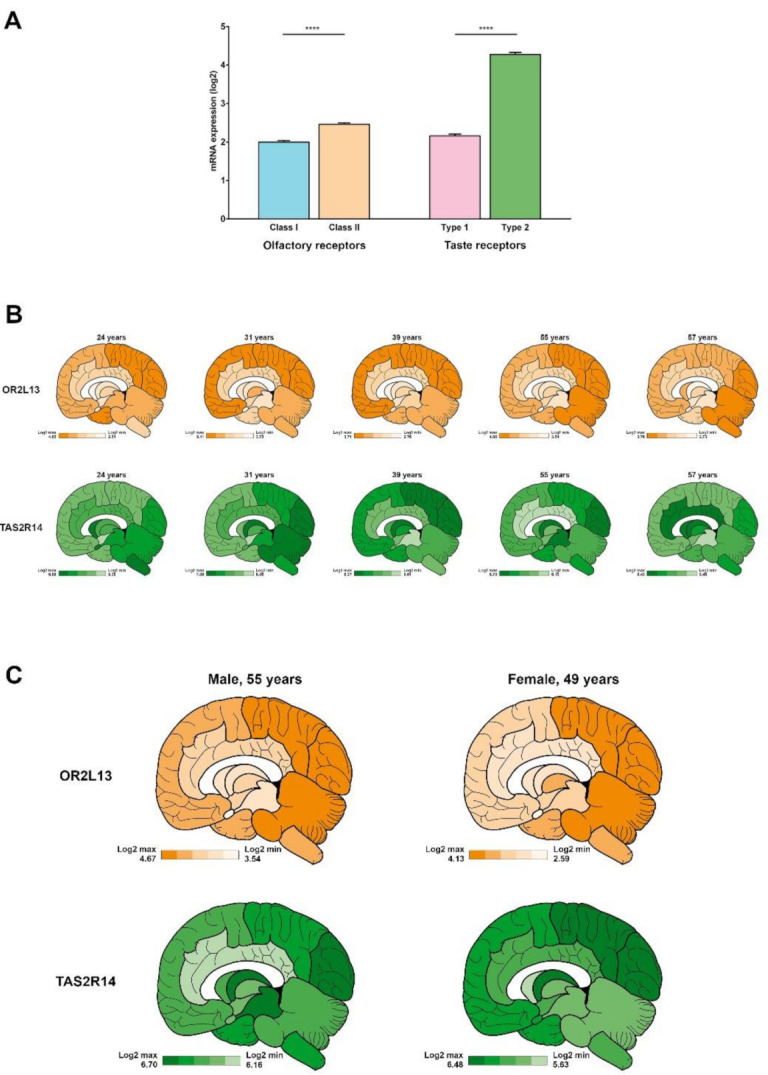
Comparative expression of olfactory and taste receptor transcripts in the human brain, and inter-individual variations in transcript expression. (**A**) global expressions of class I, class II ORs, and type 1 and type 2 TASRs in the whole brain of a 31-year-old Caucasian male. ****: *p* < 0.0001 (non-parametric Mann–Whitney test). (**B**) Five subjects of various ages were compared. OR2L13 (**top**), one of the most expressed class II ORs in the human brain, is predominantly expressed in the frontal, parietal and occipital cortices but less expressed in regions of the limbic system such as the amygdala, basal ganglia, or hypothalamus, in all subjects. In contrast, the expression pattern of TAS2R14 (**bottom**) varies a lot between subjects (**bottom**). Maximum and minimum expression values are indicated in the bottom left panel for each representation. (**C**) Comparative expression and regionalisation of the OR2L13 (**top**) and TAS2R14 (**bottom**) genes between a male and a female subjects of comparable ages. Maximum and minimum expression values are indicated in the bottom left panel for each representation.

**Figure 3 ijms-22-06858-f003:**
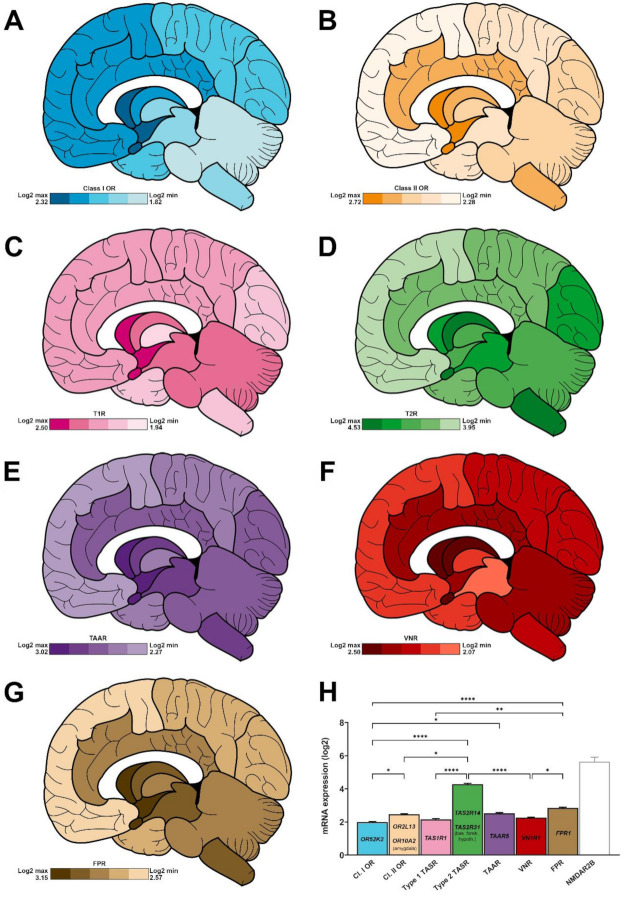
Regionalised mean expression of chemoreceptors in a human brain collected from a 31-year-old Caucasian subject. (**A**–**G**), log2 expression for class I (**A**, blue) and class II (**B**, orange) olfactory receptors, type 1 (**C**, pink), and type 2 (**D**, green) taste receptors, trace amine-associated receptors (**E**, purple), vomeronasal receptors (**F**, red), and formyl peptide receptors (**G**, brown). The maximum and minimum expression levels are indicated in the bottom left panel for each family representation. (**H**) Histogram showing the average expression for each family in the whole brain (n = 13 main areas). The most expressed members for each family is indicated inside each bar. NMDAR2B gene expression (empty bar, grey, was added as a positive control). *: *p* < 0.05, **: *p* < 0.01, ****: *p* < 0.0001 (Kruskal–Wallis test followed by Dunn’s multiple comparisons post hoc test). For clarity purposes, NMDAR2B was excluded from the statistical analysis.

**Figure 4 ijms-22-06858-f004:**
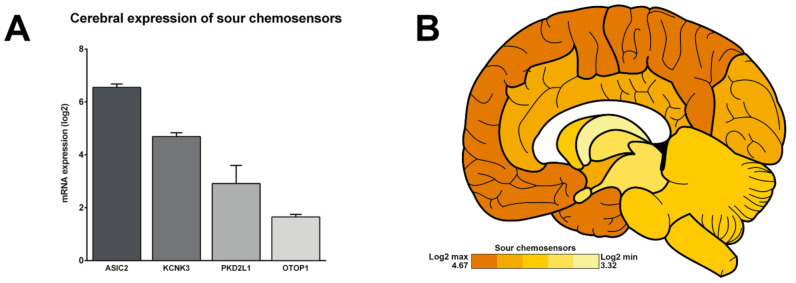
Expression of sour chemoreceptor genes in the human brain. (**A**) Histogram showing the average expression of ASIC2, KCNK3, PKD2L1 and OTOP1 in the whole brain of a 31-year-old Caucasian male. (**B**) Regionalised mean expressions of described sour chemosensor candidates in the brain of the same subject. The maximum and minimum log2 values are indicated in the bottom left panel.

**Figure 5 ijms-22-06858-f005:**
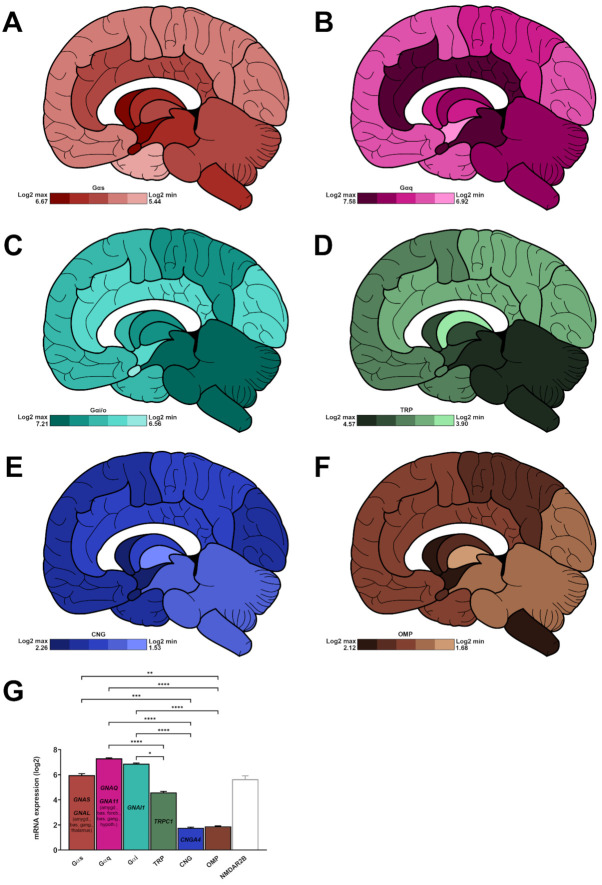
Regionalised mean expression of chemoreceptor-associated partners in a human brain collected from a 31-year-old Caucasian subject. (**A**–**F**), log2 expression for Gαs (**A**, red), Gαq (**B**, pink), Gαi/o (**C**, light blue), transient receptor potential channels (**D**, green), cyclic nucleotide-gated channels (**E**, dark blue), and olfactory marker protein (**F**, brown). The maximum and minimum expression levels are indicated in the bottom left panel for each family. (**G**) mean expression for each family in the whole brain (*n* = 13 main areas). The most expressed members for each family is indicated inside each bar. NMDAR2B gene expression (empty bar, grey, was added as a positive control). *: *p* < 0.05, **: *p* < 0.01, ***: *p* < 0.001, ****: *p* < 0.0001 (Kruskal–Wallis test followed by Dunn’s multiple comparisons post hoc test). For clarity purposes, NMDAR2B was excluded from the statistical analysis.

**Figure 6 ijms-22-06858-f006:**
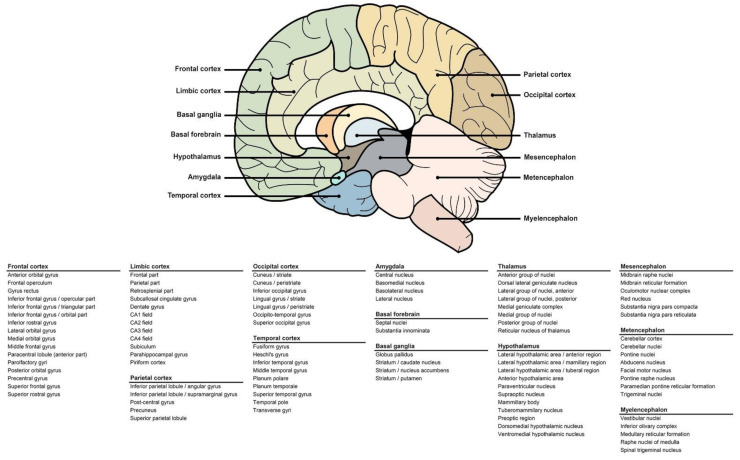
Areas of the human brain included in the study. Visual representation of the human brain with arrows indicating the main areas (**top**). Table detailing the substructures included in each main zone (**bottom**).

**Table 1 ijms-22-06858-t001:**
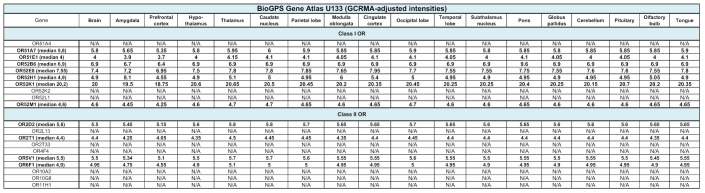
Expression of class I and class II olfactory receptor genes in other databases. N/A, not available.

**Table 2 ijms-22-06858-t002:**
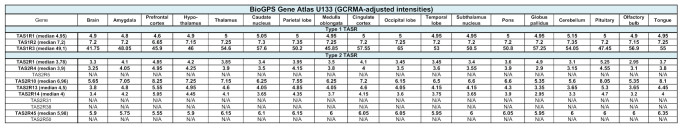
Expression of type 1 and type 2 taste receptor genes in other databases. N/A, not available.

**Table 3 ijms-22-06858-t003:** Selected olfactory receptors (OR) and taste receptors (TASR).

Olfactory Receptors		
Class I	Cerebral Areas	References
OR51A4		
OR51A7		
OR51E1	EC, PFC, FC, SN, Cb	[25,26,27,28]
OR52B6		
OR52E8		
OR52H1	EC, PFC, FC, Cb	[27]
OR52K1	Homolog of Olfr544, found in cortex and hippocampus	[29]
OR52K2	Homolog of Olfr544, found in cortex and hippocampus	[29]
OR52L1	EC, PFC, FC, SN, Cb	[25,26,27,28]
OR52M1	EC, PFC, FC, Cb	[27]
**Class II**	**Cerebral areas**	**References**
OR2D2	EC, PFC, FC, SN, Cb	[25,26,27,28]
OR2L13	EC, PFC, FC, SN, Cb	[25,26,27,28]
OR2T1	EC, PFC, FC, Cb	[27]
OR2T33	EC, PFC, FC, SN, Cb	[25,26,27,28]
OR4F4	EC, PFC, FC, SN, Cb	[26,27,28]
OR5V1	Homolog of Olfr110/111, found in cortex and hippocampus	[26,29]
OR6F1	EC, PFC, FC, Cb	[27]
OR10A2		
OR10G8	EC, PFC, FC, SN, Cb	[26,27,28]
OR11H1	EC, PFC, FC, SN, Cb	[25,26,28]
**Taste Receptors**		
**Type 1**	**Cerebral areas**	**References**
TAS1R1		
TAS1R2		
TAS1R3		
Type 2	Cerebral areas	References
TAS2R1		
TAS2R4	EC, PFC, FC, SN, Cb	[25,26,28]
TAS2R5	EC, PFC, FC, SN, Cb	[25,26,28]
TAS2R10	EC, PFC, FC, SN, Cb	[25,26,28]
TAS2R13	EC, PFC, FC, SN, Cb	[25,26,28]
TAS2R14	EC, PFC, FC, SN, Cb	[25,26,28]
TAS2R31		
TAS2R38		
TAS2R45		
TAS2R50	EC, PFC, FC, SN, Cb	[25,26,28]

The human OR family includes 396 members, clustered into class I, “fish-like”, and class II, “tetrapod-like”, while the TASR family is sub-divided into “sweet and umami” type 1 (TAS1R, 3 members), and “bitter” type 2 (TAS2R, 25 members). Large families were restricted to 10 genes. EC: entorhinal cortex, PFC: prefrontal cortex, FC: frontal cortex, SN: *substantia nigra*, Cb: cerebellum.

## Data Availability

Data used in this study were obtained from the Allen Brain Institute (http://human.brain-map.org), the BioGPS Gene Atlas (http://biogps.org/), GTEx (https://gtexportal.org/home/) and Human Protein Atlas (https://www.proteinatlas.org).

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
