# Peer review of "Limbic Expression of mRNA Coding for Chemoreceptors in Human Brain—Lessons from Brain Atlases"

_ijms, 2021, doi:10.3390/ijms22136858_

Round 1
Reviewer 1 Report
Limbic expression of mRNA coding for chemoreceptors in human brain – Lessons from brain atlases
Submitted by Dr. Fanny Gaude
Summary
- The authors performed a study mapping chemoreceptors within the encephalon based on four brain atlases.
- The authors analyzed the transcript expression of selected members of the three chemoreceptor families and their canonical partners in significant areas of healthy and demented human brains.
- Authors found that genes encoding all studied chemoreceptors are transcribed in the central nervous system, particularly in the limbic system.
- The authors also observed the RNA of their canonical transduction partners (G proteins, ion channels) in all studied brain areas. Based on this observation, they suggest cerebral chemoreceptors may be functional.
- Authors suggested that i) bitterness-associated receptors display an enriched expression, ii) the brain is equipped to sense trace amines and pheromonal cues, and iii) chemoreceptor RNA expression varies with age but not dementia or brain trauma.
- Authors also suggested that extensive studies are now required to understand further how the brain makes sense of endogenous.
The manuscript submitted by Dr. Gaude's group contains intriguing findings regarding chemoreceptors ectopically expressed in the brain.
However, there are several concerns to be cleared by authors and some suggestions.
Major Concerns
- The authors chose ten ORs from each class for the study. The authors should explain how they select such ORs. One may be afraid if it gives a bias since it is difficult to know which particular ORs are highly expressed. It would be appropriate to see the expression on the whole ORs.
- The authors stated that chemoreceptor RNA expression varies with age but not dementia or brain trauma. A number of publications suggested a correlation between OR expression and Alzheimer's disease. In particular, Dr. I. Ferrer's group showed that OR, ADYLC3, and Gnal mRNA expression is also altered in AD model mice. The authors should explain this discrepancy.
- The authors focused on the regionalized OR distribution in the brain of a single individual (31 yr-old male). Does it mean that all results regarding the regionalized OR distribution in the brain were obtained from one individual brain sample? Results from one individual brain do not seem sufficient to explain the regionalized OR distribution in the brain. The authors may consider enlarging the sample size (e.g., number of subjects).
Minor Concerns & Suggestions
For the expression value (Figure 3), is about 0.5 difference between the max and the min sufficient to compare the OR expression levels in various brain regions?
In addition to OMP, monitoring protein expressions such as ADCY3, ANO2, RIC8B, RTP1, and RTP2 would be beneficial to researchers interested in Canonical chemoreception transducers, as Flegel C. et al. (2013) did. Flegel C. et al. also showed that these ORs are expressed in brain tissue with NGS data and PCR experiments.
References
Neurology. 2011 Apr 12;76(15):1302-9. doi: 10.1212/WNL.0b013e3182166df5.
Neuroscience. 2013 Sep 17;248:369-82. doi: 10.1016/j.neuroscience.2013.06.034.
Mol Neurobiol. 2019 Mar;56(3):2057-2072. DOI: 10.1007/s12035-018-1196-4.
PLoS ONE 2013 8(2): e55368. doi:10.1371/journal.pone.0055368
Author Response
Reviewer #1
The manuscript submitted by Dr. Gaudel's group contains intriguing findings regarding chemoreceptors ectopically expressed in the brain. However, there are several concerns to be cleared by authors and some suggestions.
Major Concerns
- The authors chose ten ORs from each class for the study. The authors should explain how they select such ORs. One may be afraid if it gives a bias since it is difficult to know which particular ORs are highly expressed. It would be appropriate to see the expression on the whole ORs.
Response: We fully understand the referee's concern. From the very beginning, we considered mapping the 376 inventoried human olfactory receptors but, in the interest of space and reader’s comprehension, we opted for a random selection of 20 ORs. In order to comply with reviewer’s suggestions (1 and 3, see below), we drew a supplementary table in which we provide the level of expression for each olfactory receptor, at various ages. Nonetheless, to avoid an overloaded and unreadable table, we focused our attention on one specific brain area, the amygdala, as an example of inter-OR and inter-individual variations. A new paragraph has been added to the text which now reads:
In order to assess the most highly expressed olfactory receptors, we compared the number of copies in the four nuclei of the amygdala (supplementary table 1). The top seven ORs (log2 value > 7) within the amygdala are: OR2H1, OR3A1, OR4D2, OR5L2, OR10C1, OR10H2, OR13A1. However, it should be noted that some discordant values between probes are sometimes observed.
- The authors stated that chemoreceptor RNA expression varies with age but not dementia or brain trauma. A number of publications suggested a correlation between OR expression and Alzheimer's disease. In particular, Dr. I. Ferrer's group showed that OR, ADYLC3, and Gnal mRNA expression is also altered in AD model mice. The authors should explain this discrepancy.
Response: As pointed out by the referee, our data are discordant with previous studies performed by I. Ferrer’s group. Four of them were already mentioned in the original manuscript. The discussion chapter included the following sentences: “(…) previous studies indicated that the expression of some cerebral ORs and TASRs is modulated in patients with neuropathologies [37-40]. We assessed CR expression in brains of individuals with dementia or after a traumatic brain injury. Intriguingly, no significant variation in CR expression was observed. Such a discordant finding indicates that additional studies are required to enlighten this issue”. As recommended by the reviewer, we added a new sentence to the paragraph that now reads:
Such a discordant finding indicates that additional studies are required to enlighten this issue. However, it can be pointed out that the previous study on OR expression in the frontal and entorhinal cortex was performed with the PCR technique (38) while the team of the Allen brain atlas used cDNA microarrays.
- The authors focused on the regionalized OR distribution in the brain of a single individual (31 yr-old male). Does it mean that all results regarding the regionalized OR distribution in the brain were obtained from one individual brain sample? Results from one individual brain do not seem sufficient to explain the regionalized OR distribution in the brain. The authors may consider enlarging the sample size (e.g., number of subjects).
Response: In agreement with reviewer’s suggestion, we enlarged the cohort and compared the level of expression of all ORs between six individuals (supplementary table 1). The text has been modified accordingly. It now reads:
In addition, inter-individual variability was assessed by comparing the level of expression of all ORs within the amygdala of the six individuals included in the of the Allen brain atlas cohort (supplementary table 1). For the same receptor in the same nucleus, expression variations can range from simple to quadruple. However, no effect of age was observed.
Minor Concerns & Suggestions
- For the expression value (Figure 3), is about 0.5 difference between the max and the min sufficient to compare the OR expression levels in various brain regions?
Response: The referee is right. Mean comparison can be misleading. On average, the level of expression of the 10 selected class I ORs is pretty similar to the level of expression of the 10 selected class II ORs. However, as revealed by the newly added supplementary table 1, inter-individual and inter-area variations can be much more pronounced.
- In addition to OMP, monitoring protein expressions such as ADCY3, ANO2, RIC8B, RTP1, and RTP2 would be beneficial to researchers interested in Canonical chemoreception transducers, as Flegel C. et al. (2013) did. Flegel C. et al. also showed that these ORs are expressed in brain tissue with NGS data and PCR experiments.
Response: We could and we still can draw another supplementary table listing the expression of ADCY3, ANO2, RIC8B, RTP1, and RTP2 within the brain. However, our study is focused on chemoreceptors and not only olfactory receptors. Inserting such data would unbalance the entire study
Reviewer 2 Report
Brain reference atlases allowed the high-throughput searching for changes that could be connected to physiological as well as pathophysiological stages of the organism. There are several atlases and the comparison of data obtained after their study is also indicated. Authors of the manuscript entitled Limbic expression of mRNA coding for chemoreceptors in human brain – Lessons from brain atlases focused on charting the CR in CNS. The research was conducted correctly and the authors present possible further steps that are necessary to accomplish to verify their findings. I have minor suggestions:
Introduction - the Authors easily proceed from mouse models to Human samples. But there is no comments about the comparison of human and mouse physiology concerning the discussed receptors. The authors should provide the description why the research was constructed as presented in the manuscript.
References - maybe there are several ref that are not necessary? In my opinion there is too many of them
.
Author Response
Reviewer #2
Brain reference atlases allowed the high-throughput searching for changes that could be connected to physiological as well as pathophysiological stages of the organism. There are several atlases and the comparison of data obtained after their study is also indicated. Authors of the manuscript entitled Limbic expression of mRNA coding for chemoreceptors in human brain – Lessons from brain atlases focused on charting the CR in CNS. The research was conducted correctly and the authors present possible further steps that are necessary to accomplish to verify their findings. I have minor suggestions:
- Introduction - the Authors easily proceed from mouse models to Human samples. But there is no comment about the comparison of human and mouse physiology concerning the discussed receptors. The authors should provide the description why the research was constructed as presented in the manuscript.
Response:
We acknowledge that our transition from mouse to human may seem abrupt and we thank reviewer #2 for pointing this out. We have modified part of the introduction to, we hope, better explain our thought process:
In vertebrates, OR genes are well conserved and share a high percentage of homology as they are phylogenetically derived from nine common ancestor genes. For example, OR5V1 and OR52K1/OR52K2 are potential human orthologs of Olfr110/111 and Olfr544, respectively, with over 80% shared identity. The initial focus of the current study was to describe the expression of the latter within the human brain. Taking advantage of the freely accessible brain atlases, we used the transcriptome data (cDNA microarray, RNA sequencing) to expand our scope of investigation to a wider range of chemoreceptors and chart the genetic expression of human cerebral CRs.
- References - maybe there are several ref that are not necessary? In my opinion there is too many of them
Response: In our opinion, the number of references is adequate, especially for an article that is halfway between a classic research article and a review article. However, we wouldn’t mind removing a few if the referee is kind enough to identify them.
Round 2
Reviewer 1 Report
All my concerns are thoroughly answered.